# Development of Structural Insulated Panels Made from Wood-Composite Boards and Natural Rubber Foam

**DOI:** 10.3390/polym13152497

**Published:** 2021-07-28

**Authors:** Nussalin Thongcharoen, Sureurg Khongtong, Suthon Srivaro, Supanit Wisadsatorn, Tanan Chub-uppakarn, Pannipa Chaowana

**Affiliations:** 1School of Engineering and Technology, Walailak University, Nakhonsithammarat 80160, Thailand; nussalin@outlook.com (N.T.); ssuthon@wu.ac.th (S.S.); 2School of Architecture and Design, Walailak University, Nakhonsithammarat 80160, Thailand; supanit.wi@wu.ac.th; 3Faculty of Engineering, Prince of Songkhla University, Songkhla 90112, Thailand; ctanan.psu@gmail.com; 4Center of Excellence on Wood and Biomaterials, Walailak University, Nakhonsithammarat 80160, Thailand; 5Center of Excellence on Petrochemical and Materials Technology, Bangkok 10330, Thailand

**Keywords:** structural insulated panel, sandwich panel, wood-composite boards, natural rubber foam

## Abstract

An experimental study was carried out to develop and examine the properties of a new type of structural insulated panel (SIP). SIP prototypes conducted from this research consisted of insulated foam manufactured from natural rubber filled with wood particles as the core layer and three kinds of commercial wood-composite boards (plywood, cement particleboard, and fiber-cement board) as the surface layers. Polyurethane was used as an adhesive bond between the surface and the core layer. This preformed panel was placed into a clamping device and compressed until adhesive curing was achieved. The physical and mechanical properties of the SIP prototypes were consequently evaluated. The test results indicated that the types of surface layer materials played a significant effect on the SIP properties. The SIP covered with cement particleboard and fiber-cement board revealed high mechanical properties and high water resistance. The SIP prototype covered with plywood showed desirable properties (such as low density, high resistance of screw withdrawal, and low thermal transmittance). However, high water absorption and low fire resistance were drawbacks of the SIP covered with plywood. These properties should be improved.

## 1. Introduction

A structural insulated panel (SIP) is a sandwich-structured composite panel consisting of an insulating layer with a foam core covered with two surface layers of structural boards, such as metal sheets, oriented strand boards (OSB), plywood, or wood-cement boards. It is a type of frameless panel that is often used for various applications, such as with an exterior wall, roof, floor, or foundation system [1]. SIP is an engineering construction material because it provides several favorable features, such as being lightweight, high strength, environmentally friendly, and having excellent thermal and acoustic performance [2]. However, SIP has some disadvantages, such as the release of gas (classified as a carcinogen) from the surface layer formaldehyde-based adhesive binder, the inability of expanded polystyrene (EPS) as a core layer to break down in the environment after destruction, and low fire resistance. Moreover, SIP is still a relatively new material in some areas. Therefore, finding professional builders with experience in using SIP can be complicated [2].

There are two techniques for SIP fabrication. For the first one, an adhesive is applied on both surfaces of a pre-cut foam core, and then it is placed between two surface layers under pressure until the adhesive is cured. For the other technique, a foam core is introduced inside the space between two surface layers; thus, bonding between surfaces and a core is achieved when the foam is completely set. SIP built by these two methods provides both similar structural and insulation properties as building construction materials [3]. Usually, SIP is manufactured with a wide range of thicknesses and cutting sizes for the appropriate application and requirements [3]. It is then transported to the construction site, where a quick installation could be performed. Thus, SIP would increase energy efficiency and reduce wastes in building construction compared to the traditional way. Moreover, it could be promoted as an environmentally sustainable product for building construction.

However, the typical SIP’s core insulating materials are polystyrene (PS), expanded polystyrene (EPS), or polyurethane (PUR); and its derivative, such as polyisocyanurate (PIR). These materials raise an issue of sustainability because they are manufactured from petrochemical substances, which potentially damage the environment. Moreover, the rising price of these core materials, which are non-renewable resources, is also a concern. In order to comply with the policy for economic energy and to minimize the impact on the environment, the use of locally available natural and renewable materials has come to focus.

Natural rubber is an agricultural natural material consisting of isoprene polymers, with minor impurities of other organic compounds and water. The Association of Natural Rubber Producing Countries reported that Asia is the primary source of natural rubber, accounting for about 90% of output in 2019. The top three natural rubber-producing countries are Thailand, Indonesia, and Malaysia, which account for about 70% of the total natural rubber production globally. Natural rubber is used extensively in many applications and products, such as tires, gloves, balloons, boots, shoes, bags, erasers, silicone phone casings, and even pharmaceuticals supplies. Due to its high level of elasticity, high compressive resistance, excellent thermal properties, and good sound insulation [4], natural rubber may be considered as a material in building construction, including the core insulated material for SIP.

Phohchuay and Khongtong [5] manufactured and tested the properties of SIP made from a natural rubber foam as the core layer and rubberwood strands as the surface layers. They indicated that the properties of SIP significantly depended on the content of wood particles as a filler inside the foam core. The density of this product ranged from 0.41–0.55 g/cm^3^, which was lower than that of brick and concrete walls by approximately four times. They also revealed that this product showed high water resistance and high bending strength. Additionally, the value of thermal conductivity varied from 0.07 to 0.08 W/m·K, which was significantly lower than that of brick and concrete walls by sixteen times. Remarkably, the value of sound transmission lost was in the range of 35–80 dB, which is better than that of the wall materials currently used. Wisadsatorn and Phohchuay [6] reported that SIP, made from a natural rubber foam core covered with wood strand surface layers with a thickness of 6.5 cm, can be used as the self-containing panel for a two-story building wall with a height of 2.6 m without buckling failure. Therefore, SIP is compatible with the modular wall construction system. Comparing with previous research [5,6], the production of SIP in this research was done by the separated process where the prefabricated surface layers were bonded to the core layer. Hence, the manufacturing process can be divided into three processes: (1) production of the surface layers, (2) production of the core layer, and (3) agglutination between these layers by bonding. In this innovative process, the natural rubber compound can expand its volume three to four times upon heating. Therefore, thicker rubber foam is available. Moreover, the separated process is similar to the existing SIP manufacturing process. Some processes and equipment were slightly modified for this study.

This study applied a traditional manufacturing technique to develop SIP prototypes made from an insulated rubber foam core covered with commercial wood-composite boards in laboratory scale. The raw materials and SIP prototypes were tested for their physical properties, mechanical properties, thermal conductivity, and flammability. This research is expected to increase the efficient utilization of natural resources to create a new construction product.

## 2. Materials and Methods

### 2.1. Raw Materials

Three commercial wood-composite boards (i.e., plywood, cement particleboard, and fiber-cement board) were used as the surface layer in this research. The plywood was supplied by Thai Plywood Company, Ltd., Bangkok, Thailand. The cement particleboard was obtained from Viva Industries Company, Ltd., Chachoengsao, Thailand. The fiber-cement board was provided by Siam Cement Group Public Company, Ltd., Bangkok, Thailand. Their features are presented in Table 1.

According to the European Committee for Standardization [7] and Youngquist [8], boards with a density below 0.55 g/cm^3^ are defined as lightweight boards. One method to control the density of SIP is by controlling the wood particle content in the core layer [5,9]. Therefore, natural rubber sheets were pre-mixed with dry wood particles at 50 wt. % in order to make a masterbatch. The formula of rubber compounds shown in Table 2 was prepared according to the preliminary study results [9]. As illustrated in Figure 1, all components were mixed by two roll mills for 40 min. To ensure uniform component distribution, optimal mixing conditions were determined by pretests [9]. Then, it was sheeted out and kept overnight (approximately 10 h) before further processing.

In the next day, natural rubber compound sheets were cut and stacked into a mold with a dimension size of 95 × 250 × 480 mm^3^. These assemblies were then fabricated by compression molding under a pressure of 22 kg/cm^2^ and a temperature of about 140 °C. During the process, the heat was transferred to the rubber compound sheets to initiate a blowing agent to release gases. Subsequently, the compound in the compressed mold was allowed to expand to form the rubber foam. Usually, this step took approximately 50 min to develop the rubber foam with a thickness of 80 ± 2 mm, as shown in Figure 1.

Polyurethane adhesives (Model: GSP PU 902/GSP PU 902H) were supplied by GSP Products Co., Ltd., Bangkok, Thailand. Its properties are presented in Table 3.

### 2.2. Investigation of the Surface and Core Material Properties

According to standard requirements as mentioned below, three types of wood-composite boards were cut with a band saw into a specific size. All samples were equilibrated at a temperature of 20 ± 3 °C and 65 ± 5% relative humidity for two weeks before determining their properties. Five samples of each raw material were used for each type of test. The following properties of the samples were determined:Density test: The density with the specimen dimension of 100 mm × 100 mm × board thickness was measured following ASTM D1037-12 [10].Thickness swelling and water absorption test: The swelling of the specimen and water absorption properties were observed with the specimen dimension of 152 mm × 152 mm × board thickness. The specimens were immersed in water at 20 °C for 24 h following ASTM D1037-12 [10].Thermal conductivity test: The thermal conductivity of the specimen was measured from specimens with a dimension of 50 mm × 50 mm × board thickness at 25 °C using a Thermal Constant Analyzer (TPS 2500S Hot Disk, Hot Disk AB, Gothenburg, Sweden) according to the standard ISO 22007-2:2015 [11].Fire resistance test: The specimens with a dimension of 12 mm × 125 mm × board thickness were tested in a UL94 Horizontal/Vertical Flame Chamber according to ASTM D3801-10 [12].Three-point static bending test: The structural bending properties of the specimen were tested using the methods stated in ASTM D1037-12 [10], using a 10 kN Universal Testing Machine. Each test specimen had a nominal width of 50 mm. The length of each specimen was 51 mm plus 24 times the nominal thickness. Modulus of Rupture (MOR) and Modulus of Elasticity (MOE) values were calculated for each specimen.Compression test parallel to the surface: The compressive strength parallel to the surface of the board was measured following ASTM D1037-12 [10] (Method A: Laminated Specimen). Each wood composite board was cut into the test specimens which were 25 mm (width) × 102 mm (length) × board thickness. Polyurethane adhesive was mixed following the supplier’s suggestion, as present in Table 3, and spread out on a single glue surface with an adhesive content of 150 g/m^2^ using a hand brush. Two pieces of board were pressed together in a clamping device at a pressure of 3 N/mm^2^ and room temperature until the adhesive cured (approximately 24 h). After that, the specimen was test using a 150 kN Universal Testing Machine. The compressive strength (*R_c_*) and Modulus of Elasticity in compression parallel to the surface of the board (*E*) of each specimen were calculated.

The natural rubber foam was cut with a band saw into specific sizes for the requirement of each testing standard. Five samples were used for each type of test. The following properties of the samples were determined:Cellular structure dimensions of the natural rubber foam, such as cell shape and size, were determined by an optical microscope (ZEISS (Jena, Germany), Axioskop 2 MAT).Density was measured from specimens with a dimension of 50 mm × 50 mm × 80 mm following ASTM D1622-14 [13].Thermal conductivity was measured from specimens with a dimension of 50 mm × 50 mm × 10 mm at 25 °C using a Thermal Constant Analyzer (TPS 2500S Hot Disk, Hot Disk AB, Gothenburg, weden) according to the standard ISO 22007-2:2015 [11].The fire resistance test with the specimen dimension of 12 mm × 125 mm × 10 mm was performed in a UL94 Horizontal/Vertical Flame Chamber according to ASTM D635-10 [14].Flexural tests were performed following ASTM D 790-17 [15] to measure flexural strength. Each test specimen was sized at 50 mm (width) × 340 mm (length) × 80 mm (core thickness) at a test span of 290 mm. The testing was conducted using a 10 kN Universal Testing Machine. The test results obtained the flexural strength at 5% deformation. When the natural rubber foam, which is a homogeneous elastic material, was tested in flexure as a simple beam supported at two points and loaded at the midpoint, the flexural stress ($\sigma_{f}$) and the flexural strain ($\varepsilon_{f}$) were calculated for any point on the load-deflection curve.Compression tests were conducted by following ASTM D1621-16 [16]. Each test specimen was sized nominally at 50 mm × 50 mm × 80 mm. The test results obtained the compressive strength at 10% deformation. The compressive stress ($\sigma_{c}$) and the compressive strain ($\varepsilon_{c}$) were calculated for any point on the load-deflection curve.

### 2.3. Manufacture of SIP Prototypes

Figure 2 shows the manufacturing process or batch processing method of the SIP in laboratory. Polyurethane adhesive was blended following the supplier’s suggestion and spread out on a single glue surface with an adhesive content of 150 g/m^2^ using a hand brush. Two surfaces were overlaid at the top and bottom of the core layer. All three layers were placed into the clamping device at room temperature and pressed at a pressure of 3.5 N/mm^2^ until the adhesive cured, which was usually within 24 h. Figure 3 shows that the SIP prototypes with the dimension of 250 mm (width) × 480 mm (length) × 100 mm (total thickness) were manufactured based on three types of surface material.

### 2.4. Analysis of SIP Prototype Properties

In order to compare the properties of the SIP prototype with a commercial product, Expandable Polystyrene Concrete Sandwich Wall (EPSCSW) was obtained from Fintechnic Co., Ltd., Pathum Thani, Thailand. EPSCSW is one type of SIP generally used for construction [17], and it was selected and tested for its physical and mechanical properties following the same sample geometry and same test methods. Figure 4 shows a sample of EPSCSW with a thickness of 100 mm. It is made from a mixture of expandable polystyrene and mortar cement as the core material and two fiber-cement boards of 10-mm thickness as the surface layers.

As usual, the thickness of the SIP used to produce wall panels in residential buildings is 100 mm. The panel has a size of 2400 mm (length) × 600 mm (width) so that it can be easily lifted and handled during construction.

According to standard requirements as mentioned below, the SIP prototypes were cut with a band saw into the specimens and placed in a conditioning room maintained at 65% relative humidity and 20 °C for two weeks before determining their properties. Five samples were used for each type of test. The following properties of the SIP prototypes were determined:Density with the specimen dimension of 100 mm (width) × 100 mm (length) × 100 mm (panel thickness) was measured following ASTM D1037-12 [10].Thickness swelling and water absorption with the dimension of 152 mm (width) × 152 mm (length) × 100 mm (panel thickness), in which the specimens were immersed in water at 20 °C for 24 h, were determined in accordance with ASTM D1037-12 [10].Thermal transmittance (*U*) is a measure of heat flow through a material usually expressed in terms of thermal resistance (*R*). Usually, the transmittance is the inverse of resistance. Therefore, the *U* value of each panel could be expressed as the reciprocal of the summation of thermal resistances of all components (∑*R*). Hence, it was calculated from Equations (1) and (2):
(1)$$U=\frac{1}{\sum \;R}\;\mathrm{or}\;U=\frac{1}{\left(R_{Surface}+R_{Core}+R_{Surface}\right)}$$
(2)$$R=\frac{t}{\lambda }$$
where *R* is the thermal resistance of the material, *t* is the sample thickness, and *λ* is the thermal conductivity of the material.Screw withdrawal resistance was determined for the face side of the specimen with the dimension of 76 mm (width) × 102 mm (length) × 100 mm (panel thickness) in accordance to ASTM D1037-12 [10] with some modifications and by using countersunk self-drilling screws with a root diameter of 3 mm. This screw type can be used with plywood, cement particleboard, and fiber-cement board.A three-point bending test was carried out for the SIP specimens with a dimension of 50 mm (width) × 340 mm (length) × 100 mm (panel thickness) at a test span of 290 mm. The testing method was performed with some modifications to ASTM D7250-20 [18] because the specimen length was shorter than the standard requirement. The testing was conducted using a 150 kN Universal Testing Machine, as illustrated in Figure 5. The load and the respective deflection were recorded. The bending stiffness and the fracture behavior of the sample were analyzed.

Since SIP is one kind of sandwich panel, the mechanical behavior of SIP can be explained by the composite beam theory [19,20]. The stiffness value (*k*) of SIP is generally evaluated according to the following procedure and given by the composite beam theory.
(3)$$k=\frac{P_{pl}}{\delta }$$
where $P_{pl}$ is the load at the proportional limit and δ is the total deflection in the elastic range.

### 2.5. Statistic Analysis 

Data for each test were statistically analyzed. Analysis of variance (ANOVA) at 0.01 and 0.05 level of significance was used to test for significant differences in surface materials. When ANOVA indicated a significant difference among factors, a comparison of the means was carried out employing Duncan’s range test to identify which groups were significantly different from others.

## 3. Results

### 3.1. Properties of the Surface Materials

The properties of the surface materials are summarized in Table 4. The analysis of variance results and Duncan’s range comparison for the raw material properties are also shown in Table 4. According to the results, the surface materials had a statistically significant difference between each property.

The fiber-cement board had the highest density, while the plywood had the lowest density. Based on our knowledge, it is advantageous to have the surface material as light as possible because it can reduce the weight and density of SIP. Therefore, plywood can provide benefits for SIP weight and density.

Water resistance of surface materials was evaluated by observing water absorption after 24 h of water soaking. The result reveals that this property was dependent on the type of commercial wood-composite board. The fiber-cement board showed the highest water resistance, while the plywood showed the lowest resistance. The dimensional stability of wood composites was evaluated by the amount of thickness swelling after water soaking for 24 h at 20 °C. Like water absorption, the fiber-cement board showed the best dimensional stability, and the plywood showed the worst stability.

Thermal conductivity can be defined as the quantity of heat transmitted through the unit thickness of a material in a direction normal to the surface of the unit area due to the unit temperature gradient under steady-state conditions. It is an indicator to identify a material as a heat insulator [21]. The thermal conductivity of raw material depends on its density and composition [22,23]. The results show that the plywood had the lowest thermal conductivity value. 

The flame retardancy of the wood-composite boards was characterized by the UL-94 vertical burn test. Moreover, self-sustained burning time was recorded and presented in Table 2. The cement particleboard and fiber-cement board were classified as a V-0 rating material or an inflammable material because they did not ignite. Plywood was categorized as a flammable material because it could self-extinguish within 14.8 s after removal of the burner; furthermore, the flame did not propagate up to the holding clamp, and no flaming drips were observed. Typically, the surface layers should provide fire resistance to the SIP. Consequently, the SIP covered with cement particleboard or fiber-cement board had good fire performance ratings.

The required strength properties were generally the bending properties, measured as MOR and MOE, which significantly influenced SIP strength. Figure 6 presents the selected load versus deflection curves, which reveal a bi-linear characteristic. The failure mode observed from all specimens was the tension break at the bottom part of the specimens. The results reveal that the plywood showed the highest MOR, while the cement particleboard showed the highest MOE. Interestingly, the plywood had higher strength-to-weight and strength-to-thickness ratios than the others. This could be due to its structure, which was composed of wood veneers. Therefore, fiber orientation took place even more in the longitudinal direction of the board during fabrication, and this was consequently responsible for the increase of the bending strength. Concerning the binder, the MOE value of the two inorganic-bonded composites was better than that of the plywood. The reason is because the cement particleboard and the fiber-cement board were made by blending wood flakes or wood fibers with Portland cement in the presence of water and by allowing the Portland cement to cure to make the stiff composites have high MOE values. It is also an essential issue that the cement particleboard had a higher MOE value than the fiber-cement board. It is well known that the chemical morphology of cement and wood influences the interfacial bond strength between wood and cement. Previous studies [24,25] have reported that wood extractives such as low molecular water-soluble polysaccharides, starches, sugars, and acids inhibit or delay the cement setting. This phenomenon reduces the strength of wood-cement composites. The fiber-cement board was made of cement mixed with fibers manufactured from the refining process with high pressure, and temperature caused wood degradation, whereby the wood extractives increased [26]. For this reason, the MOE value of the fiber-cement board was lower than that of the cement particleboard.

The information on compressive strength and Modulus of Elasticity in compression parallel to the surface of the board is required when three kinds of boards are used as a load-bearing wall. Statistical analysis confirmed that the compressive strength was not affected by the board type, while the Modulus of Elasticity in compression parallel to the surface of the board was affected by the board type. The fiber-cement board and the cement particleboard were stiffer than the plywood, as presented in Table 4. The selected load-deformation curves obtained in the test displayed significant nonlinearity, as shown in Figure 7. The crushing failure mode was presented, as indicated in Figure 8.

Cement particleboard and fiber-cement board are made from a mixture of wood flakes or fibers, Portland cement, water, and small amounts of chemical additives to help the curing process. These materials do not burn and have high water resistance. They are construction materials with high durability, high dimensional stability, and high warping resistance. However, the disadvantages are the relative heaviness in weight and high thermal conductivity of these products compared with plywood manufactured from rotary peeled veneers bonded with thermosetting adhesive. By covering the SIP with plywood, the results reveal that the density of the SIP could be reduced. However, high water absorption and low fire resistance appeared to be the main disadvantages of plywood. Thus, plywood is suitable for interior applications. Additionally, it must be sufficiently protected from fire by adding fire-retardant agents.

### 3.2. Properties of the Core Material

Figure 9 shows the microstructure of the core material, which mainly consisted of porous natural rubber filled with wood particles. Visual observations revealed that it was composed of closed cells, contributing to the effects of low water absorption and the low thermal conductivity values of this foam, as discussed below. The diameter of a cell was approximately 0.044 ± 0.012 mm.

Density is one of the most critical parameters of insulating foam because it impacts foam performance for many applications, such as strength, modulus, and thermal conductivity.

Generally, materials with low thermal conductivity are used as thermal insulation. Table 5 reveals that the thermal conductivity of the natural rubber foam filled with rubberwood particles (density of 0.52 g/cm^3^) was 0.09 W/m·K, which was quite similar to that of renewable-insulating materials, such as low density fiberboard and wood wool [21]. However, the thermal conductivity value of the natural rubber foam was three times higher than that of the insulating materials manufactured from petrochemical substances (such as PS, EPS, PUR, and PIR) [27]. Even though the thermal conductivity value of the natural rubber foam made from natural and renewable materials was higher than that of petrochemical materials, the thermal conductivity value of 0.09 W/m·K was still low enough to be considered an excellent thermal insulator.

When tested under atmospheric conditions, the rubber foam core showed a low flammability resistance with a burning rate average value of 38.21 mm/min (see Table 3). Jiao et al. [27] and Petrova et al. [28] suggested that the use of fire retardants (such as hydrated alumina, magnesium hydroxide, antinomy oxide, borate or borax) could reduce the flammability of rubber products.

Flexural strength is an indication of the natural rubber foam’s stiffness when bent on a three-point apparatus. Figure 10 presents the selected stress versus strain curve, which revealed a linear behavior characteristic of the natural rubber foam. The results suggest that all specimens did not break but underwent large deflection during testing. Therefore, the load at 5% deformation was reported as flexural strength. The average value was 0.05 N/mm^2^. Compared with EPS having the flexural strength value in the range of 0.075–3.17 N/mm^2^ [29], the flexural strength of the natural rubber foam was lower.

Shear modulus is one of the important properties for measuring the stiffness of a material. In this study, the shear modulus of the natural rubber foam was calculated from the experimental value obtained from the three-point bending test by assuming that the natural rubber foam was an isotropic material. By substituting the Modulus of Elasticity of the natural rubber foam, the shear modulus of the natural rubber foam was 0.60 N/mm^2^.

Compressive resistance at 10% deformation expresses the stiffness of the natural rubber foam under a compressive force. Figure 11 shows that the curve of the stress against the strain revealed a linear-nonlinear behavior, where the crushing failure mode was also presented. The compressive resistance value of the natural rubber foam was in the same range as EPS, which was in the range of 0.07–0.3 N/mm^2^ [29].

### 3.3. Properties of the SIP Prototypes

Table 6 displays the property average values of the SIP made from natural rubber foam covered with different surface materials. The results were statistically analyzed. The values of those for the commercial product (Expandable Polystyrene Concrete Sandwich Wall; EPSCSW) were also provided for comparison.

The results show that the density values of the SIP ranged from 0.53 to 0.56 g/cm^3^. Statistical analysis confirmed that the different surface materials had a significant effect on the density of the SIP. The results illustrate that the SIP made from the natural rubber foam core covered with the plywood had the lowest density value. This could be due to the lowest density value of plywood, as seen in Table 4. The density of the SIP covered with the plywood was likely in the same range as the commercial product, as shown in Table 6. Moreover, it was lower than that of the maximum requirement of the density of lightweight panels, which is 0.55 g/cm^3^ [7,8]. Additionally, the density of the SIP was much lower than that of current wall-building materials, such as brick and lightweight concrete. In general, the use of lightweight materials for building construction would likely open up good opportunities for reducing construction and transportation costs.

Water resistance of the SIP was evaluated by observing water absorption after water soaking for 24 h. The water absorption value ranged from 7.72% to 11.20%. The dimensional stability of the SIP was evaluated by examining the thickness swelling after 24 h of water soaking. It was found that the thickness swelling values varied from 0.93% to 3.08%. These properties were also dependent on the surface material. The results show that the SIP covered with the cement particleboard and fiber-cement board exhibited high resistance to water absorption and thickness swelling, while the SIP covered with the plywood showed low resistance. The SIP made from natural rubber foam covered with the cement particleboard and the fiber-cement board showed similar water absorption and thickness swelling properties as EPSCSW, as shown in Table 6. From an application point of view, the SIP covered with the cement particleboard and the fiber-cement board could be used as an exterior wall, while the SIP covered with the plywood could be used as an interior wall.

Thermal transmittance is the heat transfer rate through one square meter of a material, which is divided by the difference in temperature across the thickness. It is expressed in watts per square meter-kelvin (W/m^2^·K). In this research, the thermal transmittance value was calculated from the thermal conductivity of the surface and the core layers. The value ranged from 0.57 to 0.84 W/m^2^·K. As presented in Table 4, the results suggested that the type of surface layer significantly influenced thermal transmittance. The SIP covered with the plywood showed the lowest value. This result might explain the low SIP thermal transmittance value, as plywood has the lowest thermal conductivity value, as illustrated in Table 4. Moreover, all of the SIP prototypes showed lower thermal transmittance values than EPSCSW.

The results demonstrate that the type of surface layer affected the screw withdrawal resistance value of the SIP. The prototype covered with the plywood showed the best resistance on screw withdrawal. Moreover, all prototypes displayed this property greater than EPSCSW, as illustrated in Table 6.

Figure 12a–d show photographs of the SIP and EPSCSW sections under the central load after failure during the three-point bending test. Also, the corresponding load-deflection curves are included in the figures. It should be mentioned that for each case, five specimens were tested; however, only one curve (the one that was the closest curve to the average) out of five curves was selected and presented in the figures. The failure mode observed from the SIP covered with the plywood demonstrated debonding of the top interface starting from the edge and then was immediately followed by breakage of the upper surface (Figure 12a). This observation is consistent with Yang et al. [30], who examined the structural properties of SIP made from plywood as the surface layers and Styrofoam as the core layer. It showed a short linear relationship between load and deflection in the first stage, followed by ascending non-linear behavior up to a peak load and then a descending branch to failure. The failure mode of the SIP covered with the cement particleboard and the fiber-cement board was face yielding with a small crack in the top layer, as shown in Figure 12b,c. The results suggest that the samples had significant post-failure strength as the core was progressively crushed. This mode occurred due to the core crushing in the mid-span loading zone. For EPSCSW, failure was due to face wrinkling under the indenter. The failure occurred in linearity in the load-deflection curve before failure and a sharp drop in load at failure, as illustrated in Figure 12d. In agreement with this study, the concrete sandwich panels manufactured from the expanded polystyrene foam as core layers had primarily shown the failure mode in face wrinkling [17]. Testing results show that the SIP prototypes failed most commonly in compression. It was assumed that the compressive strength of the surface layer can maximize the flexural strength of the SIP prototypes.

The total deflection of the SIP in the elasticity range consisted of the bending deflection (δ_b_) and the shear deflection (δ_s_) [19]. It was assumed that shear deflection in the face was very low, and total shear deflection was caused by the lower shear rigidity of the natural rubber foam. The total deflection can be expressed as follows:(4)$$\delta =\delta_{b}+\delta_{s}$$
where
(5)$$\delta_{b}=\frac{{\mathrm{PL}}^{3}}{48D}$$
(6)$$\delta_{s}=\frac{\mathrm{PL}}{4{\mathrm{bcG}}_{c}}$$
(7)$$D=E_{f}\frac{{\mathrm{bt}}^{3}}{6}+E_{f}\frac{{\mathrm{btd}}^{2}}{2}+E_{c}\frac{{\mathrm{bc}}^{3}}{12}$$
(8)$$G_{c}=\frac{E_{c}}{2\left(1+v\right)}$$
where P is the load at the proportional limit (N), L is the span length (mm), b is the sample width (mm), t is the sample thickness (mm), d is the distance between the center of the surface layers on both sides (mm), c is the core thickness (mm), *D* is the flexural rigidity (N-mm^2^), G_c_ is the shear modulus of the core layer (MPa), E_f_ is the Modulus of Elasticity of the face layer (MPa), E_c_ is the Modulus of Elasticity of the core layer (MPa), and v is Poisson’s ratio (≅0.3).

In this study, it was assumed that all layers were firmly bonded together. The surface material was much stiffer and thinner than the core material. All materials were isotropic materials. Dimensions of the SIP samples under center point bending are shown in Figure 13.

An important mechanical property of SIP is stiffness, which is a bending characteristic of the specimen that can be defined as the resistance of SIP against deformation in response to an applied bending force. From Equations (3)–(8), it can be seen that the stiffness value of the SIP depends on the core layer capacity (E_c_ and G_c_) and the elastic value (E_f_) of the surface layer governed by the types of surface material. Table 6 shows the variation of stiffness of the SIP prototypes and EPSCSW calculated using Equation (3). The results reveal that the type of surface layer significantly affected the stiffness of the SIP prototypes. The stiffness of the SIP covered with the cement particleboard showed the highest value, while the SIP covered with the plywood showed the lowest value. This phenomenon can be explained by the highest MOE value of the cement particleboard and the lower MOE value of the plywood, as presented in Table 4. In order to improve this property of the SIP prototype, the thicker surface layers having a higher MOE value should be used. The comparison of stiffness for the SIP prototypes and EPSCSW is shown in Table 6. The results show that EPSCSW was much stiffer than the SIP prototypes. This was mainly due to the strong and stiff core layer of EPSCSW, which was a mixture of expandable polystyrene and mortar cement. 

To summarize, this research reveals the possibility of developing SIP made from natural rubber foam as a core layer and wood-based panels as surface layers for the walls of buildings because of its performance. However, in this research, small specimens were tested to evaluate SIP properties affected by the type of surface layer in laboratory scale. It is well known that the strength of materials decreases when the size of the materials is increased [31]. In order to make a conclusion on the strength of the SIP prototypes for possible use in construction, the size effect of specimens influencing the mechanical properties of the SIP must be considered, particularly the panel length. A numerical analysis might be used to estimate SIP behavior, mainly compression and bending stiffness under transverse loads, concerning its size. Also, full-scale tests on SIP behavior are necessary to ensure that the assumptions on material behavior are correct or need to be adjusted.

According to this study, the production cost of the SIP prototype was approximately 40% greater than that of EPSCSW. Although foam core panels are not a new invention, the SIP made from natural rubber foam core is an innovation in this area. Since the production of the SIP prototypes took place in the laboratory, which did not allow for continuous production and mass production, this led to a cost-intensive production. However, the results reveal that the natural rubber foam appeared to be able to replace petroleum-based insulators. Moreover, natural rubber is a locally available natural and renewable material that minimizes environmental impact. In addition to the environmental benefits of the more sustainable SIP prototypes, they can significantly increase the efficiency and value of natural rubber as the material in building construction, including the core insulated material for SIP.

## 4. Conclusions

This study presents the properties of SIP made from natural rubber foam covered with different surface materials with the following conclusions.

SIP with a density ranging from 0.53 to 0.56 g/cm^3^ can be efficiently made from natural rubber foam core sandwiched between wood-based panels.Surface materials play a significant role in the properties of the SIP prototypes created in this study. The SIP covered with the cement particleboard and fiber cement board revealed high mechanical properties and high water resistance. The SIP prototype covered with the plywood showed desirable properties (such as low density, high resistance of screw withdrawal, and low thermal transmittance).SIPs made from wood-composite boards and natural rubber foam core could be alternative eco-efficient materials used in construction.

## 5. Recommendations

In order to improve the properties of SIP, seven topics should be considered in further studies. The topics are as follows: Some components of SIP, particularly those constructed with plywood and natural rubber foam core, do not have sufficient fire resistance. Buildings constructed with SIPs may put occupants at a high risk of burns and smoke inhalation. The use of cost-effective fire retardants to increase fire resistance can minimize this problem.Moisture, mold, and rot problems with SIP panels can occur, particularly if using surfaces made of plywood. It can be improved using panels with waterproof surfaces, like fiber-cement board or cement particleboard, as mentioned before. Moreover, SIPs should be protected from moisture and direct exposure to weather conditions by an approved water-resistive barrier and a weather-resistive covering.Moisture content has a crucial impact that greatly affects the behavior of SIP. Analytical/numerical models should be used to predict the effects of moisture content on the strength and structural stability of SIP.Insects can be a problem for SIP because foam and plywood provide an optimal environment for insect infestation. An insecticide should be applied within the foam core and plywood to treat for insects.Since the effect of specimen size influences the mechanical properties of SIP, full-scale tests on SIP behavior, particularly bending strength, bending stiffness, and compression under transverse load, must be considered.Investigation of the structural qualification of the SIP prototypes should be conducted for ultimate and serviceability limit state design requirements.Because of the low value of stiffness of the SIP prototypes, the surface layers having a higher MOE value should be used to improve this property.

## Figures and Tables

**Figure 1 polymers-13-02497-f001:**
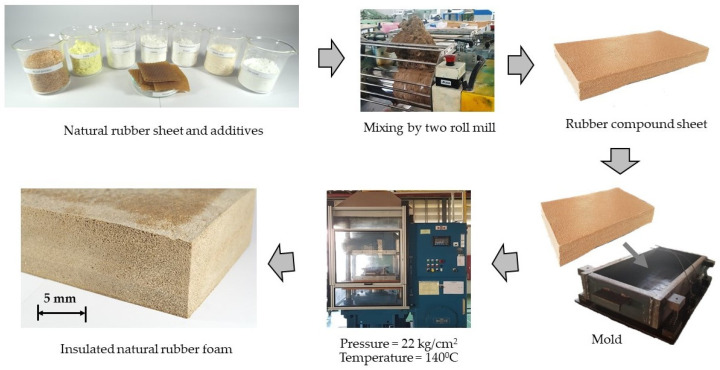
Flowchart of the manufacture of natural rubber foam in laboratory.

**Figure 2 polymers-13-02497-f002:**
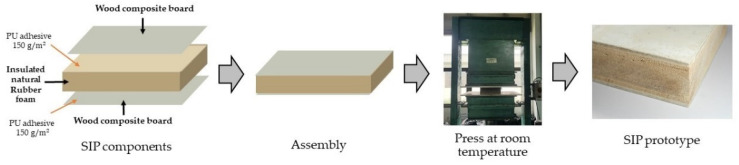
Flowchart of the manufacture of SIP prototype in laboratory.

**Figure 3 polymers-13-02497-f003:**
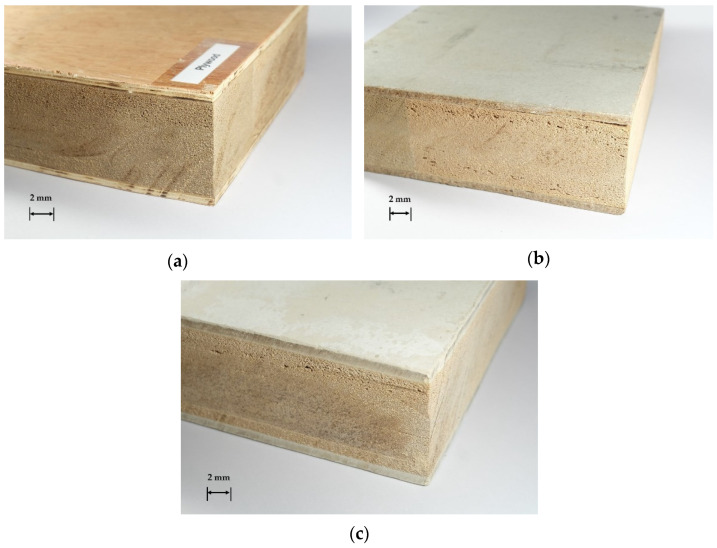
Examples of SIP prototypes made from natural rubber foam covered with (**a**) plywood, (**b**) cement particleboard, and (**c**) fiber-cement board.

**Figure 4 polymers-13-02497-f004:**
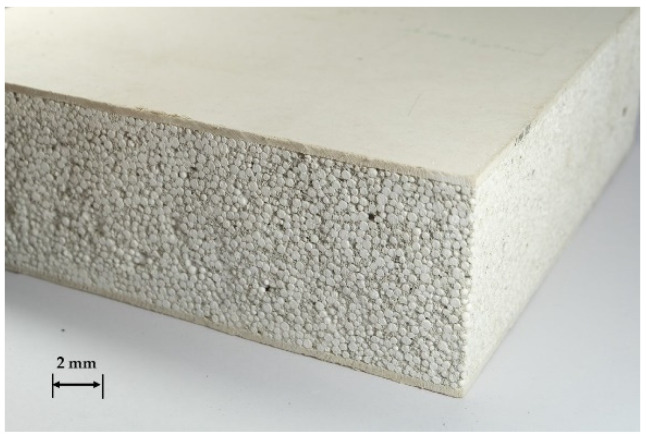
Example of Expandable Polystyrene Concrete Sandwich Wall.

**Figure 5 polymers-13-02497-f005:**
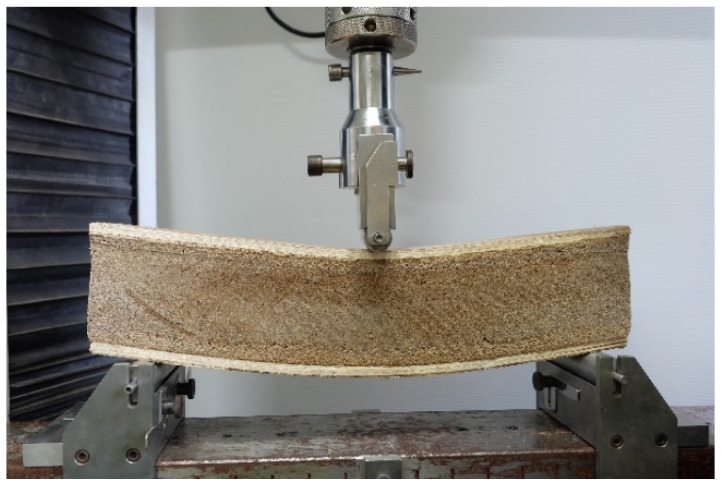
Three-point bending test of SIP prototype.

**Figure 6 polymers-13-02497-f006:**
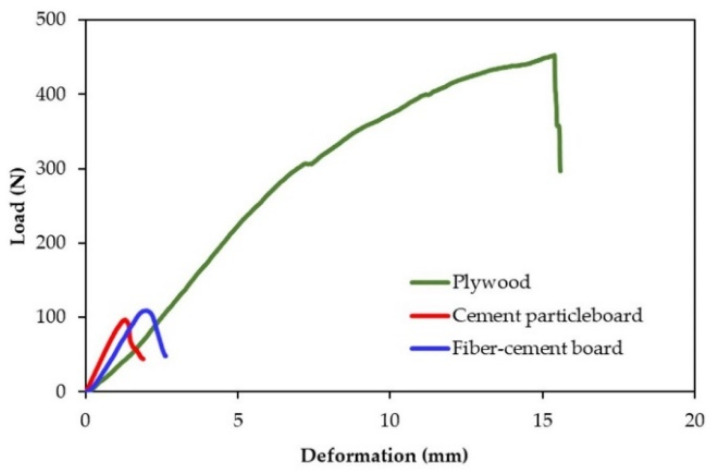
Representative load-deflection curves from the three-point bending test of the surface materials.

**Figure 7 polymers-13-02497-f007:**
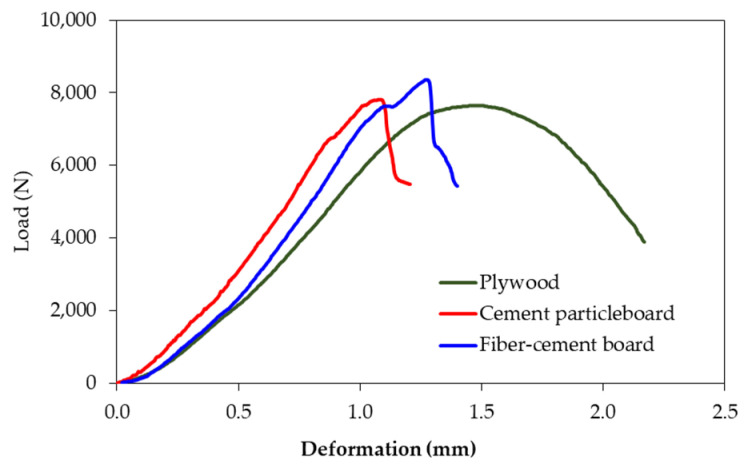
Representative load-deformation curves from the compression test parallel to the surface of the surface materials.

**Figure 8 polymers-13-02497-f008:**
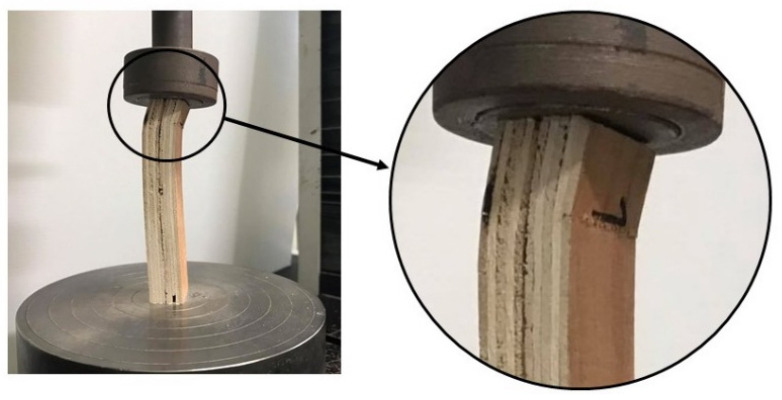
Example of crushing failure mode for compression parallel to the surface of the surface materials.

**Figure 9 polymers-13-02497-f009:**
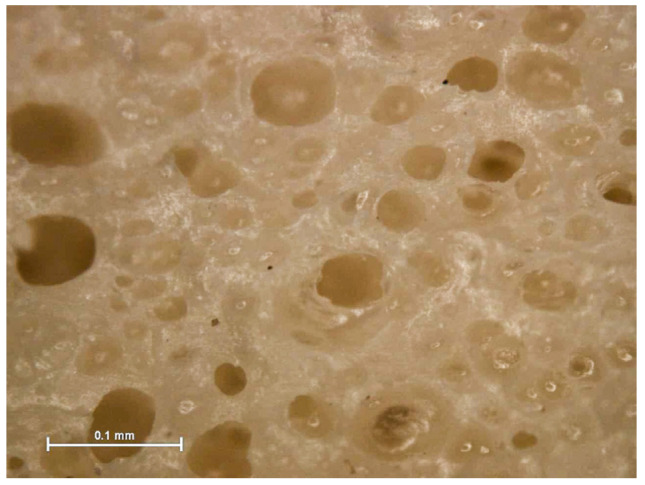
Microstructure of the porous natural rubber foam.

**Figure 10 polymers-13-02497-f010:**
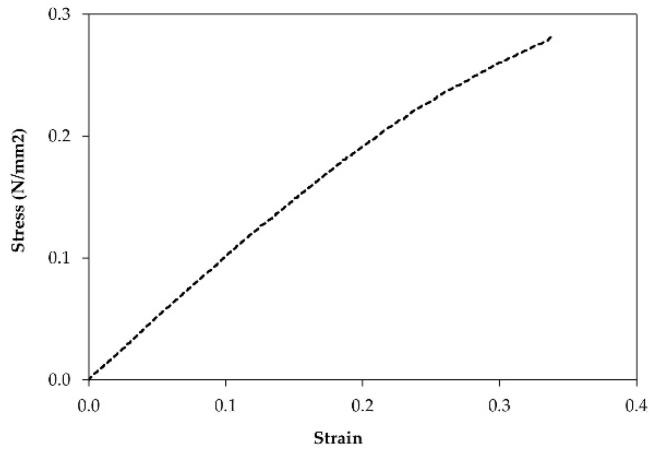
Representative stress-strain curve from the flexural test of the natural rubber foam.

**Figure 11 polymers-13-02497-f011:**
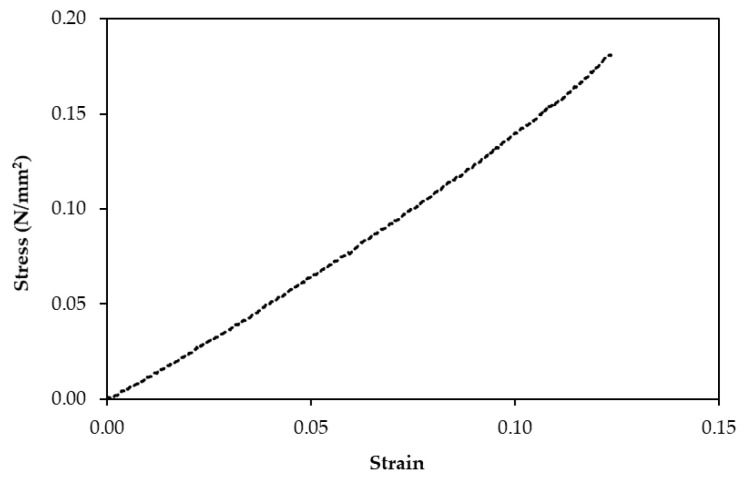
Representative stress-strain curve from the compression test of the natural rubber foam.

**Figure 12 polymers-13-02497-f012:**
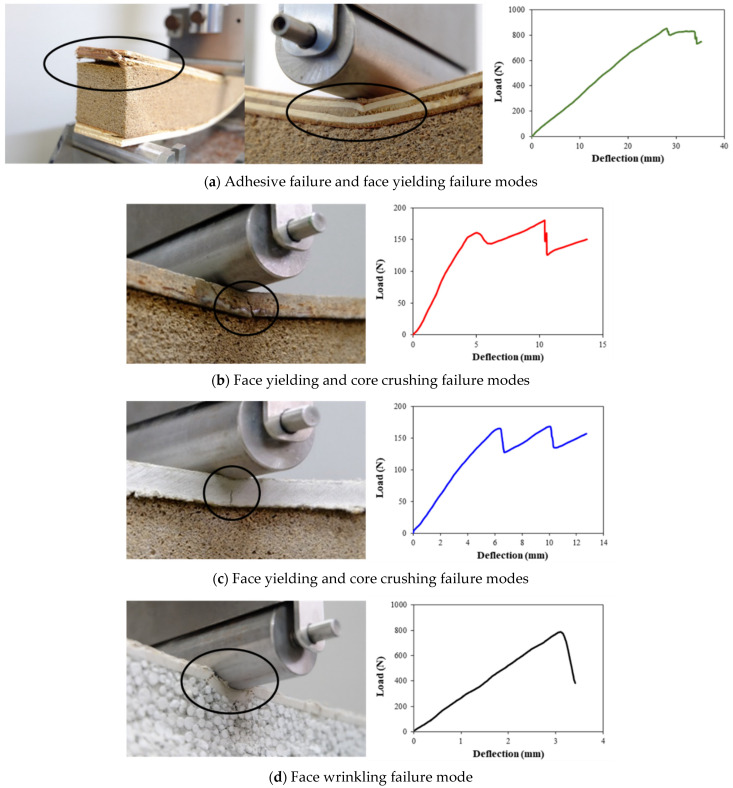
Photographs of the different failure modes for the three-point bending test of (**a**) SIP covered with plywood, (**b**) SIP covered with cement particleboard, (**c**) SIP covered with fiber-cement board, and (**d**) EPSCSW along with their corresponding representative load-deflection curves.

**Figure 13 polymers-13-02497-f013:**
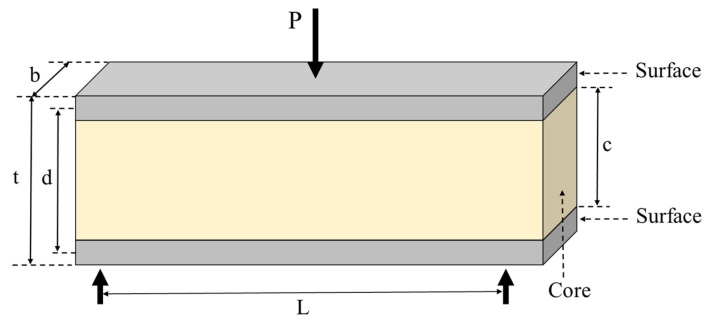
Dimensions of the SIP samples.

**Table 1 polymers-13-02497-t001:** Dimensions of the plywood, cement particleboard, and fiber-cement board used in this research.

Surface Materials	Thickness (mm)	Width (mm)	Length (mm)
Plywood	8.46	1200	2400
Cement particleboard	7.82	1200	2400
Fiber-cement board	7.62	1200	2400

**Table 2 polymers-13-02497-t002:** Formula of rubber compounds for the core layer of the SIP. The component quantity shown in the table was in the unit of part per hundred parts of rubber (phr). The functions of each component were also reported.

Components	phr	Components	Manufacturer
Natural rubber	100	Natural rubber	-
Stearic acid	2	Activator	Aldrich, St. Louis, MI, USA
Zinc oxide	2	Activator	Aldrich, St. Louis, MI, USA
Supercell-promotor	1.5	Kicker	A.F. Supercell, Bangkok, Thailand
Oxybis benzene sulfonyl hydrazide	5	Blowing agent	Merck, Kenilworth, NJ, USA
Zinc-N-diethyldithiocarbamate	1	Accelerator	Aldrich, St. Louis, MI, USA
Dibutylated Hydroxytoluene	1	Antioxidant	Aldrich, St. Louis, MI, USA
Sulfur	2	Crosslinker	Merck, Kenilworth, NJ, USA

**Table 3 polymers-13-02497-t003:** Some properties of the polyurethane adhesive used in this research.

Properties	Unit	GSP PU 902	GSP PU 902H
Chemical type	-	Polyol	Isocyanate
Appearance	-	Gray slurry	Brown liquid
Viscosity (at 30 °C)	cps.	700	200
Specific gravity (at 30 °C)	g/cm^3^	0.98	1.20
Mixing ratio	-	5	1

**Table 4 polymers-13-02497-t004:** Properties of the surface materials used in this study.

Properties	Surface Materials		
	Plywood	Cement Particleboard	Fiber-Cement Board
Density (g/cm^3^)	0.60 ^c,^* (0.01) **	1.23 ^b^ (0.04)	1.31 ^a^ (0.03)
Thickness swelling (%)	5.32 ^a^ (2.53)	2.90 ^b^ (1.12)	1.62 ^b^ (0.60)
Water absorption (%)	33.66 ^a^ (4.00)	9.21 ^b^ (0.29)	6.75 ^b^ (0.51)
Thermal conductivity (W/m·K)	0.18 ^b^ (0.00)	0.42 ^a^ (0.00)	0.43 ^a^ (0.00)
Flammability by UL-94 classification ***	V-1	V-0	V-0
Duration of self-sustained burning (Second)	14.8 ^a^ (5.26)	0.00 ^b^ (0.00)	0.00 ^b^ (0.00)
Modulus of Rupture (N/mm^2^)	32.31 ^a^ (0.97)	10.72 ^b^ (0.23)	9.93 ^b^ (0.69)
Modulus of Elasticity (N/mm^2^)	3138.42 ^c^ (189.65)	6714.63 ^a^ (139.68)	4725.12 ^b^ (258.10)
Compression strength parallel to surface (N/mm^2^)	20.87 ^a^ (2.29)	18.98 ^a^ (2.21)	20.11 ^a^ (2.08)
Modulus of Elasticity in compression parallel to surface of board (N/mm^2^)	2165.45 ^b^ (116.51)	2443.93 ^a^ (230.94)	2544.98 ^a^ (394.43)

Remarks: * Means with different superscript letters are significantly different based on grouping information using Duncan’s multiple range test (*p* < 0.05.); ** Values in parentheses represent standard deviation. *** UL-94 classification: V-0 = Material was tested in a vertical position, and it did not burn or was barely ignited; V-1 = Material was tested in a vertical position, and the burning was not sustained.

**Table 5 polymers-13-02497-t005:** Properties of the natural rubber foam used in this study.

Properties	Core Material (Natural Rubber Foam)
Density (g/cm^3^)	0.53 (0.03) *
Thermal conductivity (W/m·K)	0.09 (0.00)
Flammability by UL-94 classification **	HB
Burning rate (mm/min)	38.21 (4.15)
Flexural strength at 5% deformation (N/mm^2^)	0.05 (0.001)
Compressive strength at 10% deformation (N/mm^2^)	0.14 (0.01)

Remark: * Values in parentheses represent standard deviation. ** UL-94 classification; HB = Material was tested in a horizontal position and found to burn at a rate less than the specified maximum.

**Table 6 polymers-13-02497-t006:** Properties of the SIPs made from the wood composite boards and natural rubber foam.

Properties	SIPs Made from Natural Rubber for Core Covered with			Expandable Polystyrene Concrete Sandwich Wall
	Plywood	Cement Particleboard	Fiber-Cement Board	
Density (g/cm^3^)	0.53 ^b,^* (0.01) **	0.56 ^a^ (0.01)	0.56 ^a^ (0.01)	0.53 (0.00)
Thickness swelling (%)	3.08 ^a^ (0.17)	0.93 ^b^ (0.04)	0.93 ^b^ (0.02)	1.05 (0.03)
Water absorption (%)	11.20 ^a^ (0.41)	7.72 ^c^ (0.39)	10.26 ^b^ (0.37)	7.31 (0.20)
Thermal transmittance (W/m^2^·K)	0.57	0.83	0.84	1.70
Screw withdrawal resistance (N)	1023.03 ^a^ (96.07)	739.62 ^b^ (107.25)	480.45 ^c^ (111.97)	410.50 (51.93)
Stiffness (N/mm)	33.41 ^c^ (1.13)	41.66 ^a^ (2.59)	36.54 ^b^ (1.10)	283.69 (24.03)

Remarks: * Means with different superscript letters are significantly different based on grouping information using Duncan’s multiple range test (*p* < 0.05); ** Values in parentheses represent standard deviation.

## Data Availability

Data are contained within the article.

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
