# Peer review of "Development of Structural Insulated Panels Made from Wood-Composite Boards and Natural Rubber Foam"

_polymers, 2021, doi:10.3390/polym13152497_

Round 1

Reviewer 1 Report

Dear Authors,

I am attaching the review file.

In general, the article is interesting, but the part concerning the research methodology needs to be supplemented.

I have suggested the work to be re-written and reviewed.

Yours sincerely,

Reviewer

Reviewer 2 Report

The work is very interesting, well written with very rich experimental results and suitable to be published in the present form. But the following remarks must be considered before final acceptance.

  1. My more important concern is the type of tests performed in the work in order to study this SIP. The authors present only basic bending and compressive tests. According to me, I Think that it is insufficient to justified the use of this structures. Why shear tests, real fracture mechanics tests, the real impact of moisture content of the durability of this structure, the creep tests, are not performed? The authors must bring responses to this remarks in the conclusion for instance.
  2. Please confirm if the failure mode posted in figure 9 is for all samples.
  3. Why a numerical computation and analytical example are not proposed in this paper in order to anticipate the mechanical behavior or the fracture process of SIP?
  4. I see that the real impact on moisture content has not been investigated in the paper. The authors must discussed also in the conclusion the impact of moisture content on the efficiency of the proposed technique. Perhaps, the upcoming analytical/numerical models must be considered this important parameter in the behavior of SIP.

Author Response

Pease finds the attached file.

Reviewer 3 Report

In this contribution, Thongcharoen et al. presented a method of producing Structural Insulated Panel (SIP) using sustainable material like natural rubber and wood. Based on the presented data and explanation, I don’t feel that this manuscript is suitable to publish this journal. This manuscript contains interesting concept and a number of data supported by brief discussion, however, the work (data and scientific discussion) doesn’t present much insight in new discovery in polymer field other than using natural polymer as a starting material. I would recommend to resubmit this work in more appropriate journal like Construction Materials (MDPI).

My suggestions are given below for consideration for any resubmission:

- The manuscript is not concise and I feel that there are some details which can be revised and/or moved to the Supplementary section. Please see below.

- Fig 1 to 10 should be presented in the Supplementary information. Authors have used ASTM/ISO methods and most these photos/figures are not adding any new information on top of standard sample preparation and test procedure.

- There is not enough information on the source/manufacturer of all the materials used. Similarly information are missing on the instruments used for characterizations. Please include.

- In section 2.2, there are a lot of information repeated when detailing the tests conducted for the board and foam and SIP (section 2.4). Please make it concise.    

- There are a number of standard equations presented in this manuscript. These could be presented in the Supplementary section or removed from the manuscript.  

- For any photograph, please use scale bar.

Round 2

Reviewer 3 Report

Acceptance is recommended.

Author Response

Dear Reviewer

We are very grateful for the reviewer’s comments and have addressed all points raised by the reviewer.
